# A Thorough Examination on Zero-shot Dense Retrieval

**Ruiyang Ren**[1,3*]   **Yingqi Qu**[2]   **Jing Liu**[2†]   **Wayne Xin Zhao**[1,3†]   **Qifei Wu**[2]
**Yuchen Ding**[2]   **Hua Wu**[2]   **Haifeng Wang**[2]   **Ji-Rong Wen**[1,3]

[1]Gaoling School of Artificial Intelligence, Renmin University of China
[2]Baidu Inc.
[3]Beijing Key Laboratory of Big Data Management and Analysis Methods
{reyon.ren, jrwen}@ruc.edu.cn, batmanfly@gmail.com
{quyingqi, liujing46, wuqifei, dingyuchen, wu_hua, wanghaifeng}@baidu.com

## Abstract

Recent years have witnessed the significant advance in dense retrieval (DR) based on powerful pre-trained language models (PLM). DR models have achieved excellent performance in several benchmark datasets, while they are shown to be not as competitive as traditional sparse retrieval models (*e.g.,* BM25) in a *zero-shot retrieval* setting. However, in the related literature, there still lacks a detailed and comprehensive study on zero-shot retrieval. In this paper, we present the first thorough examination of the zero-shot capability of DR models. We aim to identify the key factors and analyze how they affect zero-shot retrieval performance. In particular, we discuss the effect of several key factors related to source training set, analyze the potential bias from the target dataset, and review and compare existing zero-shot DR models. Our findings provide important evidence to better understand and develop zero-shot DR models.

## 1 Introduction

With the massive success of pre-trained language models (PLM) (Devlin et al., 2019; Brown et al., 2020), dense retrieval (DR) has been empowered to become an essential technique in first-stage retrieval. Instead of using sparse term-based representations, DR learns low-dimensional query and document embeddings for semantic matching, which has been shown to be competitive with or even better than sparse retrievers (*e.g.,* BM25) (Karpukhin et al., 2020).

Unlike sparse retrievers, DR models typically require training on sufficient labeled data to achieve good performance in a specific domain. However, there are always new domains (or scenarios) that need to be dealt with for an information retrieval system, where little in-domain labeled data is accessable. Therefore, it is crucial

to investigate the retrieval capabilities of DR models under the *zero-shot* setting, where DR models trained on an existing domain (called *source domain*) is directly applied to another domain (called *target domain*) with no available training data on the target domain. Considering the large discrepancy between different domains, the zero-shot capability directly affects the widespread deployment of DR models in real-world applications.

Recently, a number of studies have been conducted to analyze the zero-shot capability of DR models (Thakur et al., 2021b; Chen et al., 2022), and report that DR models have poor zero-shot retrieval performance when compared with lexical models such as BM25. However, existing works primarily focus on zero-shot performance across a wide range of tasks, and lacks a comprehensive and in-depth analysis on the influence of different settings affected by various factors. Thus, the derived findings may not hold under some specific settings, which cannot ensure an accurate understanding of the zero-shot retrieval capacity. For example, our empirical analysis has revealed that BM25 is not the absolute leader compared with DR models under the zero-shot setting, and different source training sets result in different zero-shot performances (Section 2.3). As a result, a thorough examination of the zero-shot capacity of DR models is necessary.

Considering these issues, this paper aims to provide an in-depth empirical analysis of zero-shot DR. Specifically, we would like to conduct a more detailed analysis by analyzing the effect of different factors on retrieval performance. Since the DR models are trained on the source domain data (called *source training set*), we focus on the analysis by examining three major aspects of the source training set, including query set, document set, and data scale. Besides, we also consider other possible factors such as query type distribution, vocabulary overlap between source and target sets, and

---

* The work was done during the internship at Baidu.
† Corresponding authors.

the bias of the target dataset. Our analysis is particularly interested in answering the research questions of what influence factors affect the zero-shot capabilities of DR models and how does each of the influence factors affect such capability?

In addition, we also systematically review the recent advances in zero-shot DR. We summarize and categorize the key techniques that can improve the zero-shot performance (with an associated discussion of the potential influence factors).

By our empirical analysis, we find that: (i) Training data from source domain has important effect on the zero-shot capability of the trained DR model, either the query set or document set in it (Section 3.1 and Section 3.2). (ii) Vocabulary overlap and query type distribution are potential factors that affect the zero-shot performance (Section 3.1.2 and Section 3.1.3). (iii) Increasing the query scale of source data can improve both the in-domain and out-of-domain performance of the DR model (Section 3.3.1). (iv) Increasing the document scale of the source domain may result in performance decrease (Section 3.3.2). (v) The lexical bias (overlap coefficient of queries and annotated documents) of some datasets is probably one of the reasons that the performance of sparse retriever seems to be more robust than DR model on existing benchmarks (Section 3.4). Our study provides an important work to understand and improve the zero-shot capacity of the DR model.

## 2 Background and Setup

### 2.1 Zero-shot Dense Retrieval

We study the task of finding relevant documents with respect to a query from a large document set[1]. In particular, we focus on *dense retrieval* (DR) (Zhao et al., 2022) that learn low-dimensional semantic embeddings for both queries and documents and then measure the embedding similarities as the relevance scores. To implement DR, a PLM-based dual-encoder has been widely adopted (Karpukhin et al., 2020; Xiong et al., 2020; Zhou et al., 2023) by learning two separate encoders for relevance matching. Compared with sparse retrievers (*e.g.,* BM25), DR models highly rely on large-scale high-quality training data to achieve good performance (Qu et al., 2021; Ren et al., 2021a; Zhou et al., 2022).

---

[1]Note that in this paper, "document" is a generalized concept that can be in a variety of text forms, such as passage, sentence, and article.

| Datasets | #Train Q | #Eval. Q | #D | Q / D Len. |
|---|---|---|---|---|
| Natural Questions (NQ) | 58,812 | 3,610 | 21,015,324 | 9.2 / 100.0 |
| MSMARCO (MM) | 502,939 | 6,980 | 8,841,823 | 6.0 / 56.9 |
| MSMARCOv2 (MMv2) | 277,144 | 8,184 | 138,364,198 | 5.9 / 47.5 |
| HotpotQA (HQA) | 72,928 | 5,901 | 21,015,324 | 20.0 / 100.0 |
| TriviaQA (TQA) | 61,688 | 7,785 | 21,015,324 | 13.7 / 100.0 |
| SearchQA (SQA) | 117,384 | 16,980 | 21,015,324 | 14.6 / 100.0 |
| NQ$_{MRQA}$ | 104,071 | 6,775 | 21,015,324 | 11.1 / 100.0 |
| FiQA-2018 | 5,500 | 648 | 57,638 | 10.8 / 132.3 |
| SciFact | 809 | 300 | 5,183 | 12.4 / 213.6 |
| SciDocs | - | 1,000 | 25,637 | 9.4 / 176.2 |
| BioASQ | 3,742 | 500 | 14,914,602 | 8.1 / 202.6 |
| Quora | - | 10,000 | 522,931 | 9.5 / 11.4 |
| ArguAna | - | 1,406 | 8,674 | 193.0 / 166.8 |

Table 1: Dataset statistics, where "Q" denotes query and "D" denotes document. The upper part lists source datasets and the lower part lists target datasets in our zero-shot experiments.

We mainly consider the setting of zero-shot DR (Thakur et al., 2021b), where the labeled data from *target domain* is only available for testing. Besides, we assume that the labeled data from a different domain (called *source domain*) can be used to train the DR models. Such a setting is slightly different from general zero-shot learning, where no labeled data is available at all. This paper aims to present a deep and comprehensive analysis on the zero-shot capacity of DR models.

### 2.2 Experimental Setup

**Datasets** In order to comprehensively study the zero-shot capability of the DR model, we collect 12 public datasets that cover multiple domains with different data characteristics. The statistics of the datasets is shown in Table 1, including source datasets and target datasets. In our experimental design, datasets are selected that allow us to conduct controlled experiments. The target datasets are the representative datasets in BEIR (Thakur et al., 2021b) benchmark with multiple domains. The detailed description can be found in Appendix A.1.

**Training Data Construction** To examine the effect of query set and document set, which are two factors in source dataset, we incorporate a combination form of <*Query set, Document set*> to denote the resources of query set and document set, where they can come from different domains or datasets. For example, "<NQ, MM>" in Table 3 denotes that queries in NQ are used to construct training data with documents in MSMARCO.

There are two types of annotation including long answer and short answer, where long answer

| Target ($\rightarrow$) | NQ | | MSMARCO | |
| Models (with source dataset) ($\downarrow$) | M@10 | R@50 | M@10 | R@50 |
|---|---|---|---|---|
| BM25 | 31.3 | 69.7 | 18.7 | 59.2 |
| ANCE (MSMARCO) | 39.9 | 78.9 | 33.0 | 79.1 |
| RocketQAv2 (MSMARCO) | 50.8 | 83.2 | **38.8** | **86.5** |
| RocketQAv2 (NQ) | **61.1** | **86.9** | 22.4 | 67.2 |

Table 2: Evaluation results of models trained on MS-MARCO and NQ.

denotes the whole relative document and short answer denotes the short span that directly answers the query. When examining the effects of query set and document set in the source dataset, we change the query set or document set respectively. In such cases, we leverage the annotated short answers to relabel the relevant documents in a new combined source dataset. Taking "<NQ, MM>" in Table 3 as an example, we use queries in NQ and select candidates containing short answers as positives from MSMARCO document set, negatives are sampled from candidates without short answers.

**Evaluation Metrics** We use MRR@10 and Recall@50 as evaluation metrics. MRR@10 calculates the averaged reciprocal of the rank of the first positive document for a set of queries. Recall@50 calculates a truncated recall value at position 50 of a retrieved list. For space constraints, we denote MRR@10 and Recall@50 by M@10 and R@50, respectively.

### 2.3 Initial Exploration

To gain a basic understanding of zero-shot retrieval capacities, we conduct analysis on two well-known public datasets, MSMARCO and NQ, considering both in-domain (*training* and *testing* on the same dataset) and out-of-domain settings (*training* and *testing* on two different datasets).

**Backbone Selection** We select RocketQAv2 (Ren et al., 2021b) as the major DR model for study, which is one of the state-of-the-art DR models. We train models on MSMARCO and NQ, and use RocketQAv2 (MSMARCO) and RocketQAv2 (NQ) denote RocketQAv2 model trained on MS-MARCO and NQ, respectively. For comparison, we adopt the ANCE (Xiong et al., 2020) and BM25 as two baselines and report their corresponding performance. From Table 2, we find that RocketQAv2 outperforms ANCE evaluating on the two datasets (trained on MSMARCO). Thus, in the following experiments, we select RocketQAv2 as the backbone model.

**Results on NQ and MSMARCO** It also can be observed in Table 2 that the two DR models are better than BM25 when evaluated on NQ and MSMARCO. Furthermore, the performance gap evaluating on MSMARCO development set between RocketQAv2 (NQ) and RocketQAv2 (MSMARCO) (38.8 vs 22.4) is greater than the performance gap on NQ test set (61.1 vs 50.8), indicating that the DR model trained on MSMARCO is stronger than that trained on NQ.

**Results on Target Datasets** We perform the zero-shot evaluation of RocketQAv2 on six target datasets, including SciDocs, SciFact, FiQA, BioASQ, Quora, and ArguAna. As shown in Table 3 (part $A$), RocketQAv2 (MSMARCO) outperforms RocketQAv2 (NQ) on most target datasets, showing that **the DR model trained on MS-MARCO has a stronger zero-shot capability**. Moreover, it can be observed that BM25 significantly outperforms DR models on SciFact and BioASQ, and is also competitive on other datasets, which is a strong zero-shot retrieval baseline.

Through the initial experiments, we can see that models trained on the two source datasets have different zero-shot performance when tested on diverse target datasets. It is difficult to directly draw concise conclusions about the zero-shot capacities of DR models, given the significant performance variations in different settings. Considering this issue, in what follows, we systematically investigate *what* factors are relevant and *how* they affect the zero-shot performance in multiple aspects.

### 3 Experimental Analysis and Findings

In this section, we conduct a detailed analysis about multiple factors of the source training set, including source query set, source document set, and data scale. Besides, we also analyze the effect of the bias from the target dataset.

### 3.1 The Effect of Source Query Set

We first analyze the effect of different source query sets by fixing the document set. The examined query sets include NQ, MSMARCO, and datasets in MRQA (Fisch et al., 2019).

#### 3.1.1 Experimental Results

First, we fix MSMARCO as the document set and use NQ queries and MSMARCO queries as two query sets to construct training data. After training the backbone models with the two constructed

| Target (→)
Source (↓) | SciDocs | | SciFact | | FiQA-2018 | | BioASQ | | Quora | | ArguAna | | Average | |
|---|---|---|---|---|---|---|---|---|---|---|---|---|---|---|
| | M@10 | R@50 | M@10 | R@50 | M@10 | R@50 | M@10 | R@50 | M@10 | R@50 | M@10 | R@50 | M@10 | R@50 |
| *A*. Initial Experiment | | | | | | | | | | | | | | |
| BM25 | 20.7 | 57.8 | **62.2** | **86.7** | 29.7 | 63.7 | **46.0** | **73.0** | 73.8 | 95.7 | 20.8 | **86.9** | 42.2 | 77.3 |
| <NQ, Wikipedia> | **24.3** | 61.2 | 40.5 | 73.7 | 23.9 | 61.6 | 24.1 | 49.4 | **76.8** | **97.8** | 9.7 | 65.9 | 33.2 | 68.3 |
| <MM, MM> | 24.1 | **63.7** | 48.0 | 77.0 | **35.1** | **71.3** | 30.9 | 55.0 | 71.4 | 97.6 | **21.6** | 85.8 | 38.5 | 75.1 |
| *B*. Effect of Source Query Set | | | | | | | | | | | | | | |
| <NQ, MM> | 20.0 | 56.0 | 33.7 | 66.7 | 17.9 | 55.1 | 20.0 | 44.8 | 72.0 | 96.7 | 11.3 | 73.7 | 29.2 | 65.5 |
| <MM, MM> | **23.7** | **61.3** | **47.1** | **76.7** | **33.5** | **68.7** | **30.1** | **51.8** | 70.0 | 69.9 | **20.1** | **83.3** | **37.4** | **68.6** |
| <TQA, Wikipedia> | 23.1 | 61.0 | 43.9 | 77.3 | 18.4 | 54.6 | **23.9** | **47.6** | 62.5 | 92.9 | 15.4 | **79.2** | 31.2 | 68.8 |
| <SQA, Wikipedia> | 22.3 | 59.9 | **44.1** | **77.7** | 19.9 | 54.8 | 20.1 | 41.4 | 66.2 | 94.7 | **15.8** | 78.9 | 31.4 | 67.9 |
| <HQA, Wikipedia> | 18.4 | 54.6 | 34.6 | 69.7 | 11.4 | 37.4 | 18.3 | 40.6 | 46.5 | 83.8 | 10.5 | 69.1 | 23.3 | 59.2 |
| <NQ$_{MRQA}$, Wikipedia> | **24.0** | **62.7** | 29.8 | 72.7 | **24.1** | **60.5** | **23.9** | 44.4 | 71.7 | 96.8 | 14.5 | 78.2 | 31.2 | **69.2** |
| *C*. Effect of Source Document Set | | | | | | | | | | | | | | |
| <NQ, Wikipedia> | **23.5** | **60.5** | **41.2** | **72.3** | **24.5** | **60.3** | **23.5** | **49.4** | **76.1** | **97.6** | 9.7 | 66.2 | **33.1** | **67.7** |
| <NQ, MM> | 20.0 | 56.0 | 33.7 | 66.7 | 17.9 | 55.1 | 20.0 | 44.8 | 72.0 | 96.7 | **11.3** | **73.7** | 29.2 | 65.5 |
| <NQ, Wikipedia+MM> | 22.0 | 58.2 | 37.6 | 70.0 | 21.0 | 56.6 | 21.7 | 45.2 | 73.8 | 96.8 | 7.8 | 58.1 | 30.7 | 64.2 |
| *D*1. Effect of Query Scale | | | | | | | | | | | | | | |
| <NQ 10%, Wikipedia> | 7.6 | 40.9 | 32.1 | 65.7 | 6.8 | 26.1 | 9.6 | 28.0 | 50.3 | 84.2 | 9.1 | 56.8 | 19.4 | 50.3 |
| <NQ 50%, Wikipedia> | 22.1 | 59.6 | **38.7** | 70.0 | 18.9 | 54.8 | 18.0 | 42.4 | 61.4 | 92.4 | **12.0** | **71.9** | 25.2 | **65.2** |
| <NQ 100%, Wikipedia> | **23.0** | **59.7** | 36.8 | **71.3** | **20.6** | **55.9** | **18.9** | **44.4** | 65.8 | 94.0 | 10.1 | 65.2 | **29.2** | 65.1 |
| <MM 10%, MM> | 23.0 | 60.2 | 43.1 | 73.7 | 30.7 | 67.8 | 27.8 | 50.8 | 79.4 | 98.6 | 19.0 | 83.1 | 37.2 | 72.4 |
| <MM 50%, MM> | **24.4** | 62.8 | 47.0 | 74.7 | 32.9 | 68.8 | 28.9 | 53.6 | 77.7 | 98.6 | 19.8 | 84.9 | 38.5 | 73.9 |
| <MM 100%, MM> | 23.4 | **62.9** | **47.6** | **77.7** | **34.0** | **69.4** | **31.8** | **55.6** | **80.1** | **99.0** | **20.7** | **85.4** | **39.6** | **75.0** |
| *D*2. Effect of Document Scale | | | | | | | | | | | | | | |
| <MMv2 1%, MMv2> | **25.6** | **64.4** | 49.2 | **79.3** | 31.9 | 69.0 | **31.9** | **55.4** | 80.6 | **98.9** | **15.2** | **76.3** | **39.1** | **73.9** |
| <MMv2 10%, MMv2> | 24.7 | 63.6 | 48.2 | 78.0 | 32.1 | 66.7 | 30.2 | 54.6 | 77.9 | 98.7 | 14.4 | 75.1 | 37.9 | 72.8 |
| <MMv2 100%, MMv2> | 24.2 | 63.2 | **49.5** | **79.3** | **32.8** | **69.3** | 30.1 | 52.8 | 77.1 | 98.6 | 14.9 | 75.8 | 38.1 | 73.2 |

Table 3: Zero-shot evaluation results on six target datasets with different settings. We use the form <*Query Set, Document Set*> to denote the query and document sets used for training.

training datasets (denoted by $M_{NQ}$ and $M_{MARCO}$), we evaluate them on six target datasets. Moreover, we collect four QA datasets in MRQA task as source datasets from similar domains, including TriviaQA, SearchQA, HotpotQA, and NQ. Since these datasets are constructed based on Wikipedia, we adopt Wikipedia as the document set to construct training data for queries. By fixing the document set, we can analyze the effect of query sets on the retrieval performance.

Table 3 (part *B*) shows the zero-shot retrieval results on target datasets. It can be observed that the overall zero-shot capability on target datasets of $M_{NQ}$ is worse than $M_{MARCO}$, which is consistent with initial experiments (Table 3, part *A*). These results further verify that the source query set actually has important effects on the zero-shot capability of the DR model.

Given the above overall analysis, we next zoom into two more detailed factors for better understanding the effect of the query set.

### 3.1.2 Query Vocabulary Overlap

Since domain similarity is critical in cross-domain task performance (Van Asch and Daelemans, 2010), we consider studying a simple yet important indicator to reflect the domain similarity, *i.e.,* vocabulary overlap between source and tar-

get query sets. Following the previous work (Gururangan et al., 2020), we consider the top $10K$ most frequent words (excluding stopwords) in each query set. For each pair of source-target query sets, we calculate the percentage of vocabulary overlap by the weighted Jaccard (Ioffe, 2010).

Figure 1 (*red lines*) present the relationship between query vocabulary overlap and zero-shot performance on six target datasets, where we sort the source datasets according to their corresponding results from Table 3 (*x*-axis). The query vocabulary overlap between source and target datasets are shown in *y*-axis.

Overall, we can observe that **there is a certain positive correlation between query vocabulary overlap and zero-shot performance**, with only a small number of inconsistent cases. It is because a larger vocabulary overlap indicates a stronger domain similarity. Similar results (*blue lines* in Figure 1) are also found for vocabulary overlap of documents across domains.

### 3.1.3 Query Type Distribution

Besides vocabulary overlap, we continue to study the effect of another important factor, *i.e.,* query type distribution, which summarizes the overall distribution of query contents. In particular, we analyze the distribution of query types for both

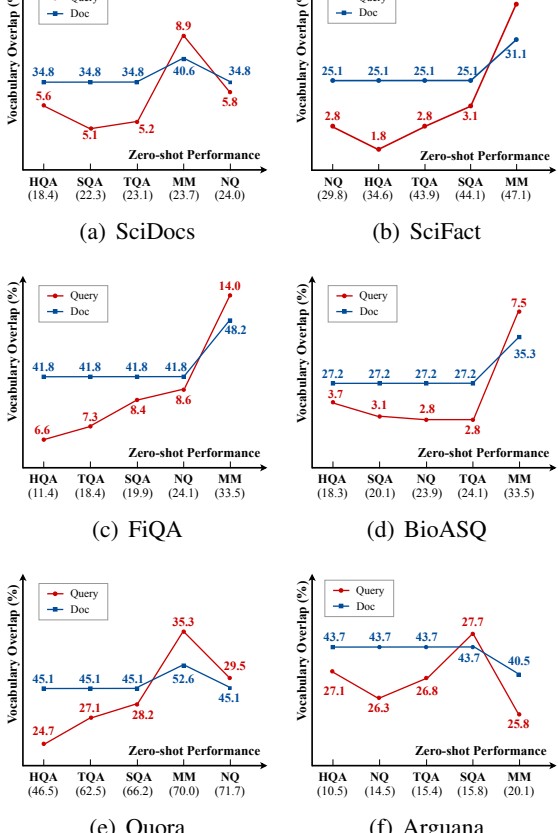

(a) SciDocs  (b) SciFact

(c) FiQA  (d) BioASQ

(e) Quora  (f) Arguana

Figure 1: Relationship between vocabulary overlap and zero-shot performance on six target datasets. The $x$-axis corresponds to the sorted zero-shot performance on the target dataset of models trained on different source datasets. $y$-axis denotes the query/document overlap of source datasets and the target dataset.

| | What | When | Who | How | Where | Why | Which | Y/N | Declarative | Entropy |
|---|---|---|---|---|---|---|---|---|---|---|
| MSMARCO | 38.7% | 2.8% | 3.8% | 13.8% | 4.4% | 1.2% | 1.6% | 7.3% | 26.3% | 1.664 |
| NQ | 11.9% | 21.3% | 40.1% | 4.2% | 9.4% | 0.2% | 1.9% | 0.3% | 10.6% | 1.648 |
| HotpotQA | 25.5% | 3.3% | 8.0% | 1.9% | 2.2% | 0.0% | 9.7% | 0.2% | 49.2% | 1.410 |
| TriviaQA | 20.6% | 0.3% | 10.9% | 1.5% | 0.8% | 0.0% | 19.9% | 0.1% | 46.0% | 1.371 |
| SearchQA | 0.1% | 0.8% | 0.0% | 0.0% | 0.0% | 0.0% | 0.0% | 0.1% | 98.9% | 0.063 |
| ArguAna | 0.1% | 0.2% | 0.3% | 0.1% | 0.2% | 0.1% | 0.0% | 0.7% | 98.2% | 0.116 |
| Quora | 37.6% | 0.9% | 2.2% | 23.7% | 1.6% | 8.8% | 4.7% | 14.6% | 5.9% | 1.707 |
| BioASQ | 32.2% | 0.8% | 0.2% | 4.0% | 0.2% | 0.0% | 17.6% | 30.6% | 14.4% | 1.504 |
| FiQA | 15.0% | 0.9% | 0.3% | 15.3% | 2.2% | 6.9% | 0.6% | 20.4% | 38.4% | 1.623 |
| SciDocs | 0.2% | 0.1% | 0.1% | 0.1% | 0.0% | 0.0% | 0.0% | 0.7% | 98.8% | 0.080 |
| SciFact | 0.0% | 0.0% | 0.0% | 0.0% | 0.0% | 0.0% | 0.0% | 0.0% | 100.0% | 0.000 |

Figure 2: Query type distribution on 11 datasets for each dataset with information entropy in last column, where top five datasets are source datasets and bottom six datasets are target datasets.

source and target query sets, focusing on "WH" queries, "Y/N" queries, and declarative queries. The query type is determined according to pre-defined rules, which are detailed in Appendix A.4.

Figure 2 presents the distribution of each query set from 11 source and target datasets. For better understanding such a distribution, we also calculate the information entropy (Shannon, 1948) of query type distribution for each dataset, where a larger entropy value indicates a more balanced distribution of query types. First, we find that **models trained with more comprehensive query types are capable of performing better in the zero-shot retrieval setting**, *e.g.,* MSMARCO dataset. In particular, MSMARCO contains the most diverse and comprehensive queries among all source datasets, which leads to a more strong zero-shot capability of DR models as the source training set.

Furthermore, **when source and target query sets share more similar query type distributions, the corresponding zero-shot performance is also likely to improve**. For example, SearchQA, containing a large proportion of declarative queries, performs better on datasets also with a large proportion of declarative queries, such as ArguAna and SciFact.

## 3.2 The Effect of Source Document Set

Following the analysis of query sets, we next study the effect of source document sets.

To conduct the analysis, we fix NQ queries as the source query set and use Wikipedia and MS-MARCO as source document sets. Furthermore, we merge the Wikipedia and MSMARCO document sets as a more comprehensive document set to study the effect of additional documents from other domain. For each query, we can use the short answer annotated in NQ dataset to select documents from three document sets to construct training data. After training with different source datasets (same query set but different document sets: Wikipedia, MSMARCO, and their combination), we evaluate the zero-shot performance of these three models on six target datasets.

Table 3 (part $C$) provides the evaluation results on six target datasets. A surprising finding is that **the model trained on the merged document set does not outperform the model trained on Wikipedia on target datasets**. A possible reason is that short answers of NQ queries annotated based on Wikipedia do not match well for other document sets, resulting in performance degradation. On the whole, the effect of the underlying document set is not that significant as query set.

| Target (→) | Query | In-domain | |
| Source (↓) | Scale | M@10 | R@50 |
|---|---|---|---|
| <NQ 10%, Wikipedia> | 5K | 40.2 | 76.1 |
| <NQ 50%, Wikipedia> | 25K | 53.4 | 84.1 |
| <NQ 100%, Wikipedia> | 50K | **56.8** | **85.0** |
| <MM 10%, MM> | 50K | 30.8 | 79.5 |
| <MM 50%, MM> | 250K | 33.8 | 83.0 |
| <MM 100%, MM> | 500K | **34.9** | **83.5** |

Table 4: In-domain experiments on NQ and MS-MARCO with different ratios of queries from the original query set.

## 3.3 The Effect of Data Scale

Data scale is also an important factor to consider that affects the zero-shot retrieval performance.

### 3.3.1 Query Scale

Query scale refers to the number of queries in the source training set. Since each training query corresponds to one (or multiple) labeled document(s) for retrieval tasks, query scale represents the amount of annotated data. We conduct experiments on NQ and MSMARCO by varying the query scale. For each dataset, we randomly sample 10%, 50%, and 100% queries from the training queries and construct three training sets. We use these training sets to train three models and conduct in-domain and out-of-domain evaluations.

Table 4 presents the results of in-domain evaluation and Table 3 (part D1) presents the out-of-domain results on six target datasets. First, **with the increase of the query scale in the training set, the in-domain capability and out-of-domain zero-shot capability of models gradually improve**. Furthermore, considering the two settings with the same query scale, *i.e.,* "NQ 100%" and "MSMARCO 10%" with 50K queries, we find the model trained on "MSMARCO 10%" still achieves a better zero-shot performance than that trained on "NQ 100%". Combining the results in Section 2.3, we find the stronger zero-shot capability of model trained on MSMARCO dataset does not simply come from the larger query scale.

### 3.3.2 Document Scale

We further analyze the effect of document scale using MSMARCOv2 (passage) dataset (Craswell et al., 2021), which is the largest publicly available text collection up to date. We randomly sample two subsets of 1% (1.4 million) and 10% (14 million) from MSMARCOv2 document set. In each setting, we fix the training queries and corresponding annotations, while the negatives are specially

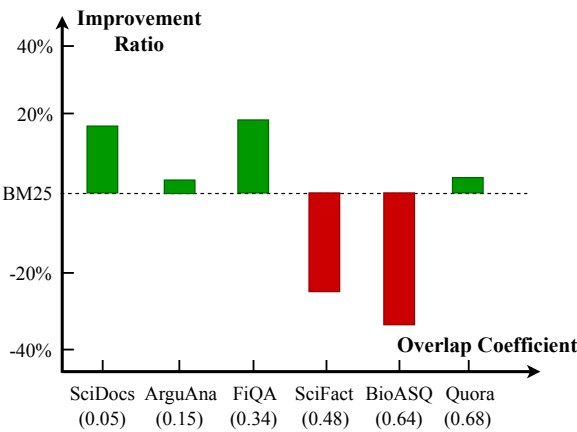

Figure 3: Relationship between the improvement ratio of DR models over BM25 and the the overlap coefficient of different target datasets.

constructed from different document sets. We then train three different models with 1%, 10%, and full document sets. More details and the in-domain evaluation results can be found in Appendix A.7.

We evaluate the models trained with different document scales on six target datasets and report the results in Table 3 (part D2). Surprisingly, the model trained on 1% document set outperforms those trained on full document set and 10% document set, which seems to contradict the intuition. We suspect that since the main discrepancies between training sets are negatives, the DR model is trained with more diverse source domain negatives using a larger document set, which absorbs more domain-specific characteristics. Therefore, **the DR model with a larger document set for training is likely to suffer from the over-fitting issue in cross-domain setting.**

## 3.4 The Bias from Target Datasets

Recently, BEIR (Thakur et al., 2021b) points out that sparse models are often used to create annotations for dataset construction, resulting in the lexical bias on the dataset, which also occurs in other tasks (Wang et al., 2022; Sun et al., 2023). According to this, we consider quantitatively investigating how such bias affects the zero-shot performance of the sparse and dense retrieval models, by computing the overlap coefficient (Vijaymeena and Kavitha, 2016) over queries and labeled documents in target test sets.

Specially, for each query from the test set of a target dataset, we compute the overlap coefficient for six target datasets, by dividing the number of overlap terms of the query and its annotated docu-

ment by the total number of query terms. We sort the overlap coefficients according to the value ascendingly. In particular, we mainly consider comparing sparse and dense retrieval models. The results on the six target datasets are shown in Figure 3. It can be observed that **BM25 overall performs better on the target dataset with a larger overlap coefficient and the performance of DR models is better than BM25 with a low overlap coefficient**. This finding indicates that the bias of the target dataset may cause it to fail to evaluate the zero-shot capabilities of different models accurately. Thus, how to construct fair datasets is also an important direction that needs to be further explored. An exception in Figure 3 is the Quora dataset. It is because that this dataset is created for duplicate question identification, where the query and the document annotation are two related questions with a larger surface similarity.

### 3.5 Summarizing the Main Findings

In this part, we summarize the main findings based on the above experimental analysis.

First, source training set has important effect on the zero-shot retrieval capacities (Section 3.1 and 3.2). Various factors related to source training set will potentially affect the zero-shot performance, including vocabulary overlap (Section 3.1.2) and query type distribution (Section 3.1.3). Our empirical study shows that it is useful to improve the zero-shot performance by increasing the vocabulary overlap and query type distribution similarity between source and target domains and setting more balanced query type distributions for the source domain.

Second, query scale of source training data significantly affects the zero-shot capability. Increasing the query scale (the amount of annotation data) brings improvement on both in-domain and out-of-domain evaluation (Section 3.3.1). However, it seems to bring negative effect when increasing the document scale, which shows a performance decrease in our experiments (Section 3.3.2).

Third, when the test set has a large overlap coefficient between queries and annotated documents, the evaluation might be biased towards exact matching methods such as BM25 (Section 3.4).

Finally, we find that MSMARCO is a more capable source dataset for training zero-shot DR models, compared with NQ. The major reason is that it contains more queries, covers more compre-

| Method | QG | KD | CP | MSS | ISR | Factors |
|---|---|---|---|---|---|---|
| QGen | ✓ | - | - | - | - | VO+QT+QS |
| AugSBERT | - | ✓ | - | - | - | QS |
| SPAR | - | ✓ | - | - | ✓ | QS |
| GPL | ✓ | ✓ | - | - | - | VO+QT+QS |
| Contriever | - | - | ✓ | - | - | QS |
| GTR | - | - | - | ✓ | - | QS |
| LaPraDoR | - | - | ✓ | - | ✓ | QS |

Table 5: Representative zero-shot DR methods with different techniques. We mark the corresponding influence factors of each methods, where VO, QT, and QS denote the improving factors of vocabulary overlap, query type and query scale, respectively.

hensive terms that may appear in other domains, and has a more balanced query type distribution.

## 4 Model Analysis

So far, we mainly focus on the discussions of data side. In this section, we further review and analyze existing zero-shot DR models.

### 4.1 Reviewing Existing Solutions

According to the specific implementations, we summarize the existing zero-shot DR models with the adopted techniques in Table 5. Next, we discuss the major techniques and the representative methods. Note that this section is not a comprehensive list of existing studies, but rather an analysis of the different techniques they use and factors from the previous analysis they improve on.

**Query Generation (QG)** QG methods construct synthetic training data by using documents from the target domain to generate (pseudo) queries, which aims to augment the training data by generating queries that fit the target domain. QGen (Ma et al., 2021) trains an Auto-encoder in the source domain to generate synthetic questions by a target domain document. Similarly, GPL (Wang et al., 2021) generates synthetic queries with a pretrained T5 model. QG can enhance the vocabulary overlap between the source and target domains, meanwhile increasing the query scale and fitting the query type distribution of the target domain.

**Knowledge Distillation (KD)** KD is a commonly used strategy in DR, which utilizes a powerful model (*e.g.,* cross-encoder) as the teacher model to improve the capabilities of DR models (Hofstätter et al., 2021). It has been found that such a technique can improve out-of-domain performance. GPL (Wang et al., 2021) and AugSBERT (Thakur et al., 2021a) use cross-encoder

to annotate unlabeled synthetic query-doc pairs to train the bi-encoder. Different from the above methods, SPAR (Chen et al., 2021) proposes to distill knowledge from BM25 to DR model. KD based approaches alleviate the data scarcity issue and play a similar role in increasing the query scale (*i.e.,* more supervision signals).

**Contrastive Pre-training (CP)** With unsupervised contrastive learning showing great power in the field of NLP, researchers start to apply this approach to zero-shot DR. Contriever (Izacard et al., 2021) and LaPraDoR (Xu et al., 2022) are two typical methods that build large-scale training pairs similar to SimCSE (Gao et al., 2021) and utilize contrastive loss in training. Such approaches use unsupervised corpus to construct training data, allowing the model to capture the matching relationship between two text contents in the pre-training phase, essentially enlarging the training data scale.

**Model Size Scaling (MSS)** For PLM based approaches, it becomes a broad consensus that scaling the model size can lead to substantial performance improvement (Brown et al., 2020; Fedus et al., 2021). Recently, MSS has shown the effectiveness in zero-shot DR. GTR (Ni et al., 2021) is a generalizable T5-based DR model that uses extremely large-scale training data and model parameters, which obtain performance gains. Moreover, large language models have attracted a lot of attention recently due to their excellent text generation capabilities (Zhao et al., 2023; Ren et al., 2023), they can also be applied to improve the zero-shot capabilities of dense retrieval.

**Integrating Sparse Retrieval (ISR)** It has been shown that both sparse and dense retrieval models have specific merits when dealing with different datasets. Thus, it is an important practical approach that integrates the two methods for enhancing the zero-shot retrieval capacities. SPAR (Chen et al., 2021) trains a student DR model that distills BM25 to the DR model, then concats the embeddings from the BM25-based DR model and a normal DR model. Moreover, LaPraDoR (Xu et al., 2022) enhances the DR model with BM25 by multiplying the BM25 score with the similarity score of DR model. We consider ISR as a special approach since it does not change the setup of source training set for DR model, but mainly combines the sparse and dense models in an ensemble way.

| Method | PLM | BioASQ | SciFact | FiQA |
|---|---|---|---|---|
| BM25 | - | 46.5 | 66.5 | 23.6 |
| QGen | DistilBERT | 39.8 | 64.4 | 30.8 |
| GTR | T5$_{base}$ | 27.1 | 60.0 | **34.9** |
| GPL | DistilBERT | 42.5 | 65.2 | 33.1 |
| LaPraDoR | DistilBERT | **51.1** | **68.7** | 34.3 |
| LaPraDoR$_{w/o\ LEDR}$ | DistilBERT | 30.8 | 59.9 | 31.4 |

Table 6: NDCG@10 results of different zero-shot methods on three target datasets.

## 4.2 Comparison of Zero-shot Methods

We compare different zero-shot methods on three most commonly used target datasets, including BioASQ, SciFact, and FiQA. Table 6 reports the performance of these zero-shot methods. We show the results of QGen reported in BEIR paper since the results in QGen paper is incomplete, and we report the T5$_{base}$ results of GTR for fairness. We replicate the results of GPL on BioASQ with the full document set, since the results in GPL paper is performed by randomly removing irrelevant documents. For LaPraDoR, we use the results of "FT" model fine-tuned on MSMARCO dataset. We also report the results without considering BM25 scores (w/o LEDR) for LaPraDoR to investigate the effect of incorporating the sparse retriever.

First, we can see that LaPraDoR overall outperforms other DR models, when remove the BM25 enhanced strategy, there is a huge drop in the zero-shot performance on BioASQ and SciFact. This corresponds to our previous analysis (Section 3.4), introducing BM25 in DR model can greatly improve the zero-shot performance on datasets with high overlap proportion (BioASQ and SciFact), while it does not bring much performance improvement on FiQA that has low overlap proportion. Without considering ISR, GPL achieves competitive performance, since it mainly fits for the vocabulary overlap and query type between the source and target sets, as well as increases the scale of pseudo labeled data (for training with knowledge distillation). Therefore, considering more positive influence factors does bring the DR model a zero-shot performance improvement.

## 5 Conclusion and Future Work

In this paper, we thoroughly examine the zero-shot capability of DR models. We conduct empirical analysis by extensively studying the effect of various factors on the retrieval performance. In particular, we find that the factors of vocabulary overlap, query type distribution, and data scale are likely

to have varying degrees of influence on the zero-shot performance of the dense retriever. Besides, the performance between BM25 and DR models varies significantly on different target datasets, where the dataset bias (*e.g.,* a dataset is created based on exact match) is likely to make such comparison unfair. Overall, we find that the zero-shot performance of dense retrieval models still has room to improve and deserves further study. As future work, we will consider designing more robust and general zero-shot retrieval methods that can adapt to different settings or domains.

## Limitations

In this work, we conduct a comprehensive empirical analysis of the zero-shot capability of the dense retrieval model, which requires large GPU resources, and we cannot conduct the same large-scale experiments on all other existing dense retrieval approaches due to the huge experimental cost and space limitation. However, since the structure of different dense retrieval models are relatively consistent, we believe that the experimental results in this paper are generalizable and can reflect the characteristics of the zero-shot capabilities of dense retrieval models.

## Acknowledgements

This work was partially supported by National Natural Science Foundation of China under Grant No. 62222215, Beijing Natural Science Foundation under Grant No. L233008 and 4222027.

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

# A Appendix

## A.1 Datasets

As mentioned in Section 2.2, we divide the collected datasets into two categories: source datasets and target datasets, corresponding to in-domain training and out-of-domain evaluation, respectively. Note that source datasets can also be used for evaluation.

### A.1.1 Source Datasets

For source datasets, we collect six famous public datasets to study various factors that may affect the transfer capability.

**Natural Questions** (Kwiatkowski et al., 2019) is a dataset for open-domain QA originally, which contains queries in Google search, documents (long answers), and answer spans from the top-ranked Wikipedia pages.

**MSMARCO** (Nguyen et al., 2016) Passage Ranking dataset contains a large number of queries with annotated passages in Web. Its queries are sampled from Bing search logs.

**MSMARCOv2** (Craswell et al., 2021) is an enhanced version of MSMARCO. It contains over 140 million passages and 11 million documents after being processed.

**HotpotQA** (Yang et al., 2018) is a question answering dataset with multi-hop questions and Wikipedia-based question-answer pairs. Multi-hop reasoning is needed to answer the questions.

**TriviaQA** (Joshi et al., 2017) is a reading comprehension dataset consists of trivia questions with correct answers and evidences from the Web.

**SearchQA** (Dunn et al., 2017) is a machine reading comprehension dataset containing question-answer pairs with snippets.

### A.1.2 Target Datasets

For target datasets, we select and reuse six datasets used in BEIR (Thakur et al., 2021b) to cover as diverse scenarios as possible, including Bio-Medical, Finance, Misc., Quora, and Scientific domains, involving Information Retrieval, Question Answering, Duplicate-Question Retrieval, Citation-Prediction and Fact Checking tasks.

**FiQA-2018** (Maia et al., 2018) is a challenge in financial domain, where opinion-based question an-

| Target (→) Source (↓) | TriviaQA | | SearchQA | | HotpotQA | | NQ | |
|---|---|---|---|---|---|---|---|---|
| | M@10 | R@50 | M@10 | R@50 | M@10 | R@50 | M@10 | R@50 |
| BM25 | 61.7 | 89.4 | 44.3 | 75.6 | 33.2 | 62.1 | 23.8 | 57.5 |
| <TQA, Wiki> | **73.4** | **93.9** | 50.7 | 80.4 | 27.7 | 54.5 | 39.0 | 70.7 |
| <SQA, Wiki> | 68.6 | 92.1 | **57.0** | **84.2** | 24.8 | 52.1 | 33.8 | 66.6 |
| <HQA, Wiki> | 64.2 | 91.3 | 47.1 | 78.1 | **34.9** | **64.4** | 36.6 | 67.7 |
| <NQ_{MRQA}, Wiki> | 63.6 | 89.6 | 44.1 | 76.4 | 25.4 | 54.4 | **50.1** | **76.7** |

Table 7: Results of models trained on four MRQA datasets, where "Wiki" denotes "Wikipedia".

swering task contains question-answer pairs from financial data.

**SciFact** (Wadden et al., 2020) is the task of scientific claim verification that selects scientific paper abstracts from the research literature to verify scitific claims.

**BioASQ** (Tsatsaronis et al., 2015) organizes challenges on biomedical semantic indexing and question answering tasks. Semantic indexing task contains queries and corresponding documents.

**SciDocs** (Cohan et al., 2020) is an evaluation benchmark for the scientific domain, consisting of seven document-level tasks, including citation prediction, document classification, and recommendation.

**Quora** (Iyer et al.) Duplicate Questions is a dataset with question pairs to identify whether two questions are duplicates that convey the same semantic information.

**ArguAna** (Wachsmuth et al., 2018) is a task of counterargument retrieval that finds the best counterargument given an argument, consisting of argument-counterargument pairs from a debate website.

## A.2 Experimental Setup

We adopted the joint training system of dense retrieval and re-ranking in RocketQAv2[2], which leverages the dynamic listwise distillation for jointly training the dual-encoder-based dense retriever and the cross-encoder-based re-ranker.

A minor modification made for the structure different from RocketQAv2 is that we add a pointwise constrain side by side with listwise constrain, since listwise score is not convenient for evaluating the relevance score of a single query-content pair. Through experiments, we find that this modification does not affect the performance of joint

---

[2] https://github.com/PaddlePaddle/RocketQA

| Target ($\rightarrow$) | NQ | | MSMARCO | |
|---|---|---|---|---|
| Source ($\downarrow$) | M@10 | R@50 | M@10 | R@50 |
| <NQ, Wikipedia> | **60.7** | **86.6** | **21.5** | **64.5** |
| <NQ, MM> | 57.8 | 86.0 | 20.3 | 63.0 |
| <NQ, Wikipedia+MM> | 59.5 | 86.4 | 19.7 | 61.3 |

Table 8: Results of DR models trained on different source document sets.

training system.

For fair comparisons, we unify the experimental settings as possible in all comparative experiments. We use the Adam optimizer (Kingma and Ba, 2015) with a learning rate of 1e-5. We run the models up to 5 epochs with a batch size of 192. All of our models are run on four NVIDIA Tesla V100 GPUs (with 32G RAM). To construct training data, we use a pre-trained DR model to retrieve the top-$k$ documents as candidates for each query in the training set. We sample undenoised instances through random sampling following RocketQAv2 and set the length of the instance list to 8 with a ratio of the positive to the negative of 1:7.

### A.3 More Results of MRQA Trained Models

For futher study, we apply models trained on four source datasets in MRQA to these four datasets with different combinations. Table 7 represents the results evaluating on MRQA datasets. The zero-shot performances of models trained on four MRQA datasets are almost all better than BM25 when evaluated on these four datasets separately, which shows that when there is not much discrepancy between the distribution of source and target domain, the zero-shot capability of dense retrieval is pretty good. We also find that the model trained on HotpotQA obtains the worst overall performance.

### A.4 Rules for Classifying Query Types

For queries starting with "WH" words, taking "what" as an example, queries with the first word of "what" or "what's" are classified as what-type queries. Queries starting with the first word "is", "was", "are", "were", "do", "does", "did", "have", "has", "had", "should", "can", "would", "could", "am", "shall". are classified as Y/N queries. The rest of the queries belong to declarative queries.

### A.5 More Results for Source Document Set Effect

We also evaluate models that are trained using Wikipedia, MSMARCO and their combination as

document sets on NQ and MSMARCO. Table 8 provides the evaluation results on NQ and MS-MARCO. We obtain similar findings as in Section 3.2 that the model trained on the merged document set does not outperform the model trained on Wikipedia on target datasets.

### A.6 Effect on Data Length

In addition to vocabulary overlap and query type, data length is a potential influencing factor on the data distribution, and we found that most of the source and target datasets do not differ much in the data length. Only special tasks such as Quora document set is shorter and ArguAna query is longer. We observe that the experimental results on these two datasets often differ from those on the other datasets (*e.g.,* Figure 1(e) and Figure 1(f)), and the specific impact of length itself on zero-shot performance is ambiguous.

### A.7 Details in Document Scale Experiments

**Training Details** During the training process, we find that training on a extreme large document set presents certain challenges. When we perform random negative sampling on entire MS-MARCOv2 document set, we analyze the training data and observe that the probability of mixing *false negatives* in it is much higher than that of the small document sets like 10% MSMARCOv2 document subset. We suspect that this is because the large document set may have more relevant contents for each topic, resulting in the serious problem of unlabeled positives.

To overcome the obstacle, we propose a data cleaning strategy for the training data from the entire document set. Concretely, we check the recall results on the 10% subset and find an average rank that roughly divides the relevant and irrelevant passages. Then use the instances on the dividing ranking position to locate a rough dividing ranking position on a large document set. We ignore the recalled documents with rank above the average dividing rank on such document set during random sampling.

**In-domain Evaluation** Table 9 presents the zero-shot retrieval results with different document scales in the source training data. It seems that the model trained on a larger document set shows better retrieval capacity for in-domain evaluation. Moreover, we find that the performance decrease of DR model is not as significant as that of BM25

| Target (→) Source (↓) | MMv2 100% | | MMv2 10% | | MMv2 1% | |
|---|---|---|---|---|---|---|
| | M@10 | R@50 | M@10 | R@50 | M@10 | R@50 |
| BM25 | 6.4 | 24.7 | 16.7 | 44.8 | 31.3 | 63.6 |
| <MMv2, MMv2 100%> | **20.6** | **63.7** | **44.0** | **86.2** | 61.7 | 93.7 |
| <MMv2, MMv2 10%> | 20.4 | 63.3 | **44.0** | 86.0 | 62.0 | 94.0 |
| <MMv2, MMv2 1%> | 16.4 | 59.1 | 40.0 | 84.8 | **62.3** | **94.5** |

Table 9: Experiments with varying document scales on MSMARCOv2.

when increasing the document scale for evaluation, which is different from the findings of previous work (Reimers and Gurevych, 2020). A possible reason is that they utilize synthetic data to construct large document set, which have a large gap with the actual scene, and cannot be well addressed by DR model.

**In-depth Analysis** Since MSMARCOv2 contains a significantly more number of documents than MSMARCO, we would like to examine whether it leads to a large expansion in vocabulary. We calculated the vocabulary overlap between MSMARCO and MSMARCOv2 document sets at different scales with the method in Section 3.1.2. We observe 99.2% overlap between "MSMARCOv2 10%" and full MSMARCOv2 document set and 93.8% overlap between "MSMARCOv2 1%" and full MSMARCOv2 document set. In addition, MSMARCO and MSMARCOv2 document sets have a overlap proportion of 79.1%. These findings show that MSMARCOv2 has a similar vocabulary size with MSMARCO, which don't incorporate substantial new topics. This may be the reason for the poor zero-shot capability of the model trained on the large document set, since model is more likely to overfit the document set.