# OpenReview forum: "A Thorough Examination on Zero-shot Dense Retrieval"
_EMNLP/2023/Conference — EMNLP 2023 Findings_

### Official Review · Reviewer_19Ji · 2023-07-28

**Typos Grammar Style And Presentation Improvements:** 1. In line 204, Furthermore, The ... .
**Soundness:** 3

**Excitement:**

3: Ambivalent: It has merits (e.g., it reports state-of-the-art results, the idea is nice), but there are key weaknesses (e.g., it describes incremental work), and it can significantly benefit from another round of revision. However, I won't object to accepting it if my co-reviewers champion it.

**Missing References:**

1. ADVERSARIAL RETRIEVER-RANKER FOR DENSE TEXT RETRIEVAL, ICLR 2022
2. ColBERTv2: Effective and Efficient Retrieval via Lightweight Late Interaction, NAACL 2022

**Paper Topic And Main Contributions:**

Since traditional sparse retrieval models (e.g., BM25) in a zero-shot retrieval setting perform better than dense retrieval (DR) model, this paper first presents a thorough examination of the zero-shot capability of DR models. This paper can help us to understand the zero-shot capacity of the DR model.

**Questions For The Authors:**

1. Is there some way to use the conclusions drawn from this paper to improve performance in zero-shot setting?

**Reasons To Accept:**

The experimental evaluation is adequate and draws some useful conclusions.

**Reasons To Reject:**

1. RocketQAv2 looks a bit dated. It would be better to adopt a newer model such as AR2 and ColBERTv2. It is not clear if these conclusions are applicable to other PLMs.
2. Some of the conclusions weren't very exciting. For example, the model trained on MSMARCO dataset achieves better performance than NQ.

**Reproducibility:**

2: Would be hard pressed to reproduce the results. The contribution depends on data that are simply not available outside the author's institution or consortium; not enough details are provided.

**Reviewer Confidence:**

1: Not my area, or paper was hard for me to understand. My evaluation is just an educated guess.

---

> ### Author Rebuttal · Authors · 2023-08-24
>
> Thanks for your insightful feedback!
>
> #Reject reason 1:
>
> To the best of our knowledge, AR2, ColBERTv2, and RocketQAv2 are studies released around the same timeframe, with the longest time gap between their release dates being less than two months.
>
> Furthermore, the inconsistency among dense retrievers generally lies in the training process, while dual-encoder architectures are typically consistent. It will not affect the trend under different factor settings with such an architecture, and we have also examined it on ANCE (a well-known dense retriever).
>
> Another advantage of using RocketQAv2 as the backbone model is that a well-trained re-ranker can also be obtained during training to easily facilitate the analysis of the re-ranker meanwhile.
>
> #Reject reason 2:
>
> We understand that not all conclusions are exciting for everyone, but also have their values. Take your example as an illustration. We find that the model trained on MSMARCO has a stronger zero-shot capability. Therefore, people who require applying existing models in new domains can give priority considering models trained on MSMARCO, where our conclusion serves as a valuable application guideline, although it is not an important finding in our analysis.
>
> Moreover, for the community, there is presently no exhaustive exploration to disclose and summarize the law of zero-shot capabilities of dense retrieval models. We believe our analysis holds a unique and valuable contribution that is beneficial to the community.
>
> #Question 1:
>
> It is a good question. Yes, besides the improvement of the existing methods mentioned in the paper can correspond to different factors, we also examined some training strategies based on our findings. For example, we attempted to generate queries using a T5-based model in the target domain to obtain a large set of queries. Then, based on the distribution of query types in the source domain, we constructed the same distribution in the training data of the target domain. The results show that controlling the query distribution led to better performance.

---

### Official Review · Reviewer_8buS · 2023-08-05

**Soundness:** 3

**Excitement:**

3: Ambivalent: It has merits (e.g., it reports state-of-the-art results, the idea is nice), but there are key weaknesses (e.g., it describes incremental work), and it can significantly benefit from another round of revision. However, I won't object to accepting it if my co-reviewers champion it.

**Paper Topic And Main Contributions:**

This paper mainly focuses on the study of dense retrieval in a zero-shot setting. Specifically, it identifies key factors influencing model performance by controlling a single variable and provides a detailed analysis of how these factors impact the performance. First, the authors analyze the impact of four variables on the DR model in a zero-shot setting, such as the source query set, source document set, data scale, and bias in the target dataset. Then, they conduct a detailed analysis of the specific factors within these variables. Additionally, the authors point out that bias in the target dataset may impact the evaluation. Finally, the authors also analyze which factors the existing technology improves performance by influencing.

**Questions For The Authors:**

Please refer to Reasons To Reject.

**Reasons To Accept:**

1. The author's experiments and analysis have addressed the previously existing gap in detailed and comprehensive research on dense retrieval in a zero-shot setting.
2. The author systematically analyzed various factors and performed corresponding experiments to illustrate the impact of different aspects of these factors on model performance. This analysis begins at a foundational level and progressively delves deeper.
3. The author's conducted experiments are robust and offer substantial evidence to substantiate the author's perspective.
4. The paper is well written and structured.


**Reasons To Reject:**

1. The author highlights that the higher the vocabulary overlap between the source training set and the target test set, the better the model's performance in a zero-shot setting. However, with an exceedingly high overlap rate, do both datasets still meet the criteria for in-domain and out-of-domain?
2. The author implemented certain allocations to control variables within the training set; however, these allocations appear to have certain issues. For instance, as mentioned in section 2.2, they utilized queries from NQ, while positive samples were extracted from the MSMARCO document set. Could this potentially result in certain queries lacking accurate answers?


**Reproducibility:**

3: Could reproduce the results with some difficulty. The settings of parameters are underspecified or subjectively determined; the training/evaluation data are not widely available.

**Reviewer Confidence:**

2: Willing to defend my evaluation, but it is fairly likely that I missed some details, didn't understand some central points, or can't be sure about the novelty of the work.

---

> ### Author Rebuttal · Authors · 2023-08-24
>
> Thanks for your insightful feedback!
>
> #Reject reason 1:
>
> It is a good question. When two datasets have an extremely high lexical overlap rate, they may be in the same domain. However, such a situation does not contradict the conclusion in our paper. Our conclusion is based on the comparison of the two overlap rates and is unrelated to their absolute values. Whether the vocabulary overlap rates are high or low, a relatively higher vocabulary overlap rate always leads to better zero-shot performance.
>
> #Reject reason 2:
>
> It is also a good question. The first thing to note is that such an issue does not impact the conclusion. It is possible that a small number of queries may not be able to find positive passages among the candidate passages. However, this type of query makes up a tiny portion, less than 1%. We discard the query when there is no positive passage, which does not hurt the performance.

---

### Official Review · Reviewer_SQPY · 2023-08-05

**Soundness:** 4

**Excitement:**

3: Ambivalent: It has merits (e.g., it reports state-of-the-art results, the idea is nice), but there are key weaknesses (e.g., it describes incremental work), and it can significantly benefit from another round of revision. However, I won't object to accepting it if my co-reviewers champion it.

**Paper Topic And Main Contributions:**

The paper conducts thoroughly investigation on the factors which might impact the zero-shot capability of dense retrieval. The paper conducts experiments on using different queries and document sets and also investigate the lexical bias of different datasets on different bias. I think the paper provides some good insights for future research. For example, query augmentation maybe the key to improve dense retrieval training and also mention the current lexical bias issue of evaluation.

**Questions For The Authors:**

1. Do you have any solution to address the current lexical bias issue of evaluation?
2. It is not very clear to me that how you construct NQ queries vs MARCO passages? If you use answer span to judge if a passage is relevant or not, there may be more than one relevant or no passages for each query. In the case of more than one relevant, you use all the passages as positive (this means you may include irrelevant passages)? In the case of no passage, you just ignore the queries?

**Reasons To Accept:**

1. The authors conduct thorough experiments on the impact of query or document for dense retrieval training
2. The conclusion paves way to training strategies, such as increasing query diversity (type and vocabulary overalp) rather than using larger documents.

**Reasons To Reject:**

1. It is pity that the authors could think about new training strategies given all the analysis rather than just reviewing existing methods.
2. Although the experiments are thorough and give some good insights, in my opinion, the paper would be more valuable one year ago before all the other pre-training dense retrieval model published. With all the recent work, the insights provide by the paper may not seem to be very excited to me. Or the authors may provide a better training strategies given all the insights as mentioned.


**Reproducibility:**

3: Could reproduce the results with some difficulty. The settings of parameters are underspecified or subjectively determined; the training/evaluation data are not widely available.

**Reviewer Confidence:**

3: Pretty sure, but there's a chance I missed something. Although I have a good feel for this area in general, I did not carefully check the paper's details, e.g., the math, experimental design, or novelty.

---

> ### Author Rebuttal · Authors · 2023-08-24
>
> Thanks for your insightful feedback!
>
>
> #Reject reason 1:
>
> This is a good suggestion.
>
> First of all, the main objective of this work is to conduct a comprehensive analysis of the factors that affect zero-shot dense retrievers and contribute reliable conclusions. There is presently no exhaustive exploration to disclose and summarize the zero-shot capabilities of dense retrievers. We believe our analysis is valuable and beneficial to the community.
>
> Actually, based on the insights of the paper, we also examined some training strategies different from existing methods. For example, we attempted to generate queries using a T5-based model in the target domain to obtain a large set of queries. Then, based on the distribution of query types in the source domain, we constructed the same distribution in the training data of the target domain. The results show that controlling the query distribution led to better performance. We are considering incorporating them in the future version.
>
> #Reject reason 2:
>
> Although there are some zero-shot dense retrievers released, most of them are designed to improve on a specific aspect. The improvements of many existing methods can correspond to the influencing factors summarized in our paper. We believe that the value of this paper lies in the systematic analysis of the influencing factors, which is a more high-level summary. Moreover, as mentioned before, we are also considering increasing the discussion of our new training strategies.
>
> #Question 1:
>
> The presence of lexical bias stems from how the dataset was annotated and is unrelated to the training method, which is a fairness issue. Thus, it is imperative to construct impartial datasets for fair comparisons in future research.
>
> #Question 2:
>
> For each query, we only use one positive passage for training. When there are multiple positive passages, we use the passage with the highest score from a ranking model following RocketQAv2. We discard the query when there is no positive passage, but this rarely happens.

---

### Meta-Review · Area_Chair_QSRi · 2023-09-19

**Recommendation:** 3

**Metareview:**

The paper investigates dense retrieval in a zero-shot setting, with a focus on identifying factors that impact model performance. It examines four key variables: source query set, source document set, data scale, and bias in the target dataset, and provides an in-depth analysis of how these variables influence model performance. The paper also highlights the potential impact of bias on evaluation. Overall, it offers a comprehensive study of dense retrieval in a zero-shot context.

There are several strengths to this paper. It fills a gap in the research by providing a detailed and systematic analysis of dense retrieval in a zero-shot setting. The experiments conducted are robust and provide substantial evidence to support the findings. The paper is well-structured and well-written, making it accessible to the reader.

However, there are a few areas of concern. Reviewers suggests that the paper could have proposed new training strategies based on the insights gained from the analysis, rather than solely reviewing existing methods. Additionally, given the recent surge in pre-training-based dense retrieval models, the insights provided by the paper may not be as groundbreaking as they would have been a year ago. To address this, the authors might consider offering more innovative training strategies based on their findings.
Furthermore, there are questions raised regarding the allocation of queries from NQ and positive samples from the MSMARCO document set. This allocation could potentially result in queries without accurate answers, which should be addressed or clarified in the paper.

---

### Decision · Program_Chairs · 2023-10-07

**Decision:**

Accept-Findings

**Comment:**

The paper investigates dense retrieval in a zero-shot setting, with a focus on identifying factors that impact model performance. It examines four key variables: source query set, source document set, data scale, and bias in the target dataset, and provides an in-depth analysis of how these variables influence model performance. The paper also highlights the potential impact of bias on evaluation. Overall, it offers a comprehensive study of dense retrieval in a zero-shot context.

There are several strengths to this paper. It fills a gap in the research by providing a detailed and systematic analysis of dense retrieval in a zero-shot setting. The experiments conducted are robust and provide substantial evidence to support the findings. The paper is well-structured and well-written, making it accessible to the reader.

However, there are a few areas of concern. Reviewers suggests that the paper could have proposed new training strategies based on the insights gained from the analysis, rather than solely reviewing existing methods. Additionally, given the recent surge in pre-training-based dense retrieval models, the insights provided by the paper may not be as groundbreaking as they would have been a year ago. To address this, the authors might consider offering more innovative training strategies based on their findings.
Furthermore, there are questions raised regarding the allocation of queries from NQ and positive samples from the MSMARCO document set. This allocation could potentially result in queries without accurate answers, which should be addressed or clarified in the paper.